# Exploring HIV-1 Maturation: A New Frontier in Antiviral Development

**DOI:** 10.3390/v16091423

**Published:** 2024-09-06

**Authors:** Aidan McGraw, Grace Hillmer, Stefania M. Medehincu, Yuta Hikichi, Sophia Gagliardi, Kedhar Narayan, Hasset Tibebe, Dacia Marquez, Lilia Mei Bose, Adleigh Keating, Coco Izumi, Kevin Peese, Samit Joshi, Mark Krystal, Kathleen L. DeCicco-Skinner, Eric O. Freed, Luca Sardo, Taisuke Izumi

**Affiliations:** 1Department Biology, College of Arts and Sciences, American University, Washington, DC 20016, USA; amcgraw@american.edu (A.M.); gh2297a@american.edu (G.H.); sm4872a@american.edu (S.M.M.); sg1978a@american.edu (S.G.); kn0119a@american.edu (K.N.); ht8146a@american.edu (H.T.); dm6732a@american.edu (D.M.); lb9210a@american.edu (L.M.B.); akeating@american.edu (A.K.); cizumi@american.edu (C.I.); decicco@american.edu (K.L.D.-S.); 2Virus-Cell Interaction Section, HIV Dynamics and Replication Program, Center for Cancer Research, National Cancer Institute, Frederick, MS 21702, USA; yuta.hikichi@nih.gov (Y.H.); efreed@mail.nih.gov (E.O.F.); 3ViiV Healthcare, 36 E. Industrial Road, Branford, CT 06405, USA; kevin.m.peese@viivhealthcare.com (K.P.) samit.r.joshi@viivhealthcare.com (S.J.); mark.r.krystal@viivhealthcare.com (M.K.); 4District of Columbia Center for AIDS Research, Washington, DC 20052, USA

**Keywords:** human immunodeficiency virus type I (HIV-1), maturation, capsid, Gag, Gag-Pol, Förster resonance energy transfer (FRET), protease inhibitor (PI), maturation inhibitor (MI), capsid inhibitor (CAI), allosteric integrase inhibitor (ALLINI)

## Abstract

HIV-1 virion maturation is an essential step in the viral replication cycle to produce infectious virus particles. Gag and Gag-Pol polyproteins are assembled at the plasma membrane of the virus-producer cells and bud from it to the extracellular compartment. The newly released progeny virions are initially immature and noninfectious. However, once the Gag polyprotein is cleaved by the viral protease in progeny virions, the mature capsid proteins assemble to form the fullerene core. This core, harboring two copies of viral genomic RNA, transforms the virion morphology into infectious virus particles. This morphological transformation is referred to as maturation. Virion maturation influences the distribution of the Env glycoprotein on the virion surface and induces conformational changes necessary for the subsequent interaction with the CD4 receptor. Several host factors, including proteins like cyclophilin A, metabolites such as IP6, and lipid rafts containing sphingomyelins, have been demonstrated to have an influence on virion maturation. This review article delves into the processes of virus maturation and Env glycoprotein recruitment, with an emphasis on the role of host cell factors and environmental conditions. Additionally, we discuss microscopic technologies for assessing virion maturation and the development of current antivirals specifically targeting this critical step in viral replication, offering long-acting therapeutic options.

## 1. Introduction

The human immunodeficiency virus (HIV) is in the genus *Lentiviridae* of the *Retroviridae* family. Currently, over 39.9 million people live with HIV worldwide, with 1.3 million new infections reported in 2023 (UNAIDS). HIV predominantly infects CD4 T-lymphocytes, inducing a rapid decline in their numbers that, if not treated, eventually leads to acquired immunodeficiency syndrome (AIDS). Effective combination antiretroviral therapy (cART) successfully halts disease progression, reducing AIDS-related deaths by 69% since the peak in 2004 and by 51% since 2010 (UNAIDS). This advancement has allowed many people living with HIV (PLWH) to achieve a normal life expectancy. While cART effectively suppresses virus replication, it does not eradicate infected cells, and HIV persists in latent reservoirs [1,2,3]. As a result, an HIV functional cure has not been achieved with the current cART, necessitating lifelong treatment for PLWH. Ongoing research aims to optimize cART regimens and develop new drugs with improved safety profiles and longer dosing intervals. The current cART regimens for HIV treatment typically consist of different classes targeting the viral enzymes: reverse transcriptase (RT), protease (PR), and integrase (IN). These classes include nucleoside and non-nucleoside reverse transcriptase inhibitors (NRTIs and NNRTIs, respectively), protease inhibitors (PIs), and integrase strand transfer inhibitors (INSTIs), often formulated into single-tablet regimens containing two or three drugs to enhance efficacy, minimize resistance, and improve patient adherence (Figure 1C,E). In addition to antiretroviral drugs targeting viral enzymes, current cART regimens may also include entry inhibitors and fusion inhibitors, which block the virion from binding to and fusing with target cells [4] (Figure 1A,B). Fusion inhibitors, in particular, belong to a class of antiretrovirals that could be used in HIV pre-exposure prophylaxis (PrEP) [5], enabling a comprehensive approach to HIV treatment and prevention. Although most current cART regimens are designed for once-daily oral dosing, recently developed long-acting antivirals are designed to be administered by injection once every few months [6]. The first and currently only long-acting regimen, Cabenuva, a combination of an INSTI (cabotegravir: CAB) and an NNRTI (rilpivirine: RPV), is administered as a once-monthly or bi-monthly HIV treatment and was approved by the FDA in 2021 [7]. Following Cabenuva, Lenacapavir (LEN) was approved by the FDA in December 2022 as a first-in-class antiviral and has the potential for applications in long-acting regimens. LEN is a capsid inhibitor (CAI), a new class of antivirals that target multiple stages of the viral replication cycle, including viral maturation (Figure 1D,H,K and Table 1) [8].

HIV-1 Gag is an essential protein involved in the formation of virus particles. It is composed of four major domains: matrix (MA), capsid (CA), nucleocapsid (NC), and p6, along with the short spacer peptides SP1 and SP2 (Figure 2A). The production of infectious virus particles is mediated by the enzymatic activity of the viral PR released from the Gag-Pol polyprotein in the progeny virion [9]. Upon the budding of progeny virions from virus-producer cells, the proteolytic processing of the Gag polyprotein by PR is initiated. This enzymatic process results in important structural rearrangements and the formation of each mature Gag protein (Figure 1K) [9]. MA plays a crucial role in the incorporation of envelope glycoproteins (Env) into the budding virions [10]. The N-terminal domain of MA in the Gag polyprotein is myristoylated and binds to the plasma membrane, which is anchored by electrostatic interactions in its highly basic region (HBR) to facilitate progeny virion assembly [11,12]. CA assembles into approximately 250 hexameric and exactly 12 pentameric units, forming the conical fullerene core that encapsulates the viral genomic RNA (gRNA). This structure is vital for reverse transcription, nuclear import, and uncoating post-infection [13]. NC is responsible for packaging the viral gRNA into the progeny virions [14,15]. NC also facilitates reverse transcription in the core upon its release to the cytoplasm post-fusion by inducing the annealing of the transfer RNA (tRNA) primer to the viral gRNA to initiate the synthesis of double-stranded viral DNA [15,16]. Lastly, p6 recruits the endosomal sorting complex required for transport (ESCRT) machinery, which is necessary for virion budding and subsequent release, and directs the incorporation of the Vpr protein into the virion [17].

In this review article, we discuss the importance of virion maturation in the viral replication cycle and how host factors affect the HIV maturation process. We also provide an overview of several classes of inhibitors targeting virion maturation, their mechanisms of action, the current state of their clinical development, and the next generation of technologies to further understand these mechanisms.

## 2. Viral Replication Cycle

HIV primarily targets CD4 T-cells by utilizing its surface protein gp120 to attach to the host cell [18]. The virus binds to the CD4 receptor and a co-receptor, either the C-C chemokine receptor type 5 (CCR5) or C-X-C chemokine receptor type 4 (CXCR4), on the T-cell surface (Figure 1A) [18]. Following virion attachment, the viral lipid envelope fuses with the target cell plasma membrane, releasing the core harboring the viral gRNA into the host cell (Figure 1B). Inside the core, RT initiates the synthesis of double-stranded viral DNA (Figure 1C) [19]. The core travels into the nucleus through the nuclear pore and uncoats upon the completion of viral DNA synthesis within the nucleus (Figure 1D,E). Eventually, the newly synthesized viral DNA integrates into the host genome, a process catalyzed by the IN complex (Figure 1E) [18,20,21]. The viral genome integrated into the DNA of a host cell is referred to as a provirus. The host RNA polymerase II transcribes the proviral DNA. This transcription is significantly enhanced by the HIV-1 Tat protein through its binding to the transactivation response (TAR) region of the viral messenger RNA (mRNA), eventually activating HIV-1 transcriptional elongation (Figure 1F) [22,23]. Tat also recruits the host transcription elongation factor b (p-TEFb) complex to the HIV-1 long terminal repeat (LTR) promoter region, increasing the efficiency of transcription and thereby promoting the production of viral RNA (vRNA) [22]. Full-length unspliced vRNA transcripts initially undergo splicing to generate fully spliced RNA species that encode solely Tat, Rev, and Nef [24,25,26,27]. These transcripts enhance the subsequent singly or partially spliced RNA species encoding Vpu: Env, and the other accessory proteins, Vif and Vpr. Some vRNA transcripts remain unspliced and serve as the viral mRNA template for the synthesis of the Gag and Gag-Pol polyproteins or as the viral genome that is packaged into nascent progeny particles (Figure 1I) [18,26,28,29]. Gag and Gag-Pol polyproteins assemble at the plasma membrane, where the immature HIV-1 particle buds from the host cell while acquiring a portion of the host cell membrane and incorporating the viral envelope (Env) glycoproteins (Figure 1G–J) [18,30]. The virus becomes infectious after maturation, a process that involves the PR-mediated cleavage of Gag and Gag-Pol polyproteins to form a mature HIV-1 core (Figure 1K) [18,31,32]. Virion maturation is the final stage in the viral replication cycle when various biochemical and morphological changes occur to ensure the transition of immature, noninfectious particles into mature, infectious viruses (Figure 1K) [32].

## 3. Molecular Mechanisms of HIV-1 Maturation

### 3.1. Fundamentals in HIV-1 Maturation

The translational ratio of Gag to Gag-Pol polyproteins from the same full-length viral mRNA is determined by a ribosomal frameshift during translation [33,34]. Approximately 5–10% of the ribosomes that initiate translation of the full-length vRNA undergo a −1 frameshift, leading to the translation of the Gag-Pol polyprotein instead of the Gag polyprotein [34,35]. While the Gag polyprotein encompasses the MA, CA, NC, and p6 domains, the Gag-Pol polyprotein lacks the p6 domain due to the ribosomal frameshift. However, the Gag-Pol polyprotein additionally includes the Pol domains, PR, RT, and IN, which are crucial for viral replication and integration into the host genome [36,37]. The Gag polyprotein undergoes cleavage in a stepwise manner, starting between SP1 and NC, followed by MA-CA and SP2-p6 cleavages, and finally removing SP1 and SP2 short peptides from CA and NC, respectively (Figure 2A) [38]. The Gag-Pol polyprotein is produced through a frameshifting event at the NC/SP1 boundary, translating the p6 domain in an alternative reading frame (Transframe/TF) [39]. The initial cleavage events separate the Gag and Pol regions by cleaving between TF and PR, a process mediated by the poorly active PR precursor (Figure 2B) [40,41]. This step releases the protease from the precursor, allowing the ordered processing of the Gag and Pol polyprotein. This proteolytic processing of the Gag and Gag-Pol polyproteins induces significant architectural changes, transforming the immature virion with a dense outer protein layer (the immature Gag lattice) into a mature virion with a defined conical core (Figure 1K and Figure 3) [42]. The mature CA assembles into a closed lattice of hexameric and pentameric units, which is essential for protecting the viral gRNA and ensuring the productive infection of new host cells (Figure 1A,B and Figure 3) [36,37].

### 3.2. Host Proteins in HIV-1 Maturation and Core Association

Several host factors play various roles in facilitating or inhibiting HIV-1 virion maturation and core formation. Cyclophilin A (CypA) is a peptidyl-prolyl isomerase that binds to the HIV-1 CA protein via a proline-rich loop, comprising CA residues 85 to 93, with glycine and proline at positions 89 and 90, respectively, serving as key binding motifs for CypA [50,51,52]. While CypA is packaged into the budding virion to support the generation of infectious viruses [17,52,53,54], its primary function is mediated by the CypA expressed in target cells, interacting with the invading HIV-1 CA core. This interaction, occurring with multiple loops on the capsid exterior in the target cell cytoplasm, influences HIV infection in a cell-type-specific manner [55,56,57,58], likely by affecting capsid stability and uncoating [59,60]. In myeloid cells, inhibiting this interaction via cyclosporin A (CsA) or mutating the CypA binding loop on CA leads to HIV-1 inducing immune responses, possibly due to premature uncoating and viral DNA release in the cytoplasm [61]. The exact molecular mechanisms remain unclear but may involve dynamic allostery [62]. Additionally, humans encode a large family of tripartite motif-containing (TRIM) proteins that recognize the CA lattice once the core is released into the cytoplasm of the target cell [63]. This recognition stimulates restriction factors targeting the viral core [64]. Although human TRIM5α has some activity against HIV-1, it is not as potent as TRIM5α from other primates, such as rhesus macaques [65]. The human TRIM5α protein has a limited ability to recognize and bind to the HIV-1 core, which is one reason why HIV-1 infection can successfully be achieved in human cells [66]. In contrast, TRIM5α from rhesus macaques can strongly restrict HIV-1 by recognizing the CA core more effectively, leading to a more robust antiviral response [67,68]. An alternative splicing form of the chimeric transcript encoding the TRIM motif fused to a C-terminal CypA domain (TRIMCyp) also shows antiviral activity by binding to the proline-rich CypA-binding loop on the HIV-1 CA [59,69,70]. However, TRIMCyp is not naturally expressed in humans [71,72]. In addition, the restriction activity facilitated by rhesus TRIM5α along with TRIMCyp also improves CD8 T-cell-mediated inhibition of HIV-1, indicating that there is a direct link between rhesus TRIM5α and CD8 T-cell response to infection [73]. The human TRIM5α may have lost these activities through evolution, or HIV-1 may have evolved to escape human TRIM5α-mediated restriction [58,73,74,75].

Cleavage and polyadenylation specificity factor 6 (CPSF6) is a cellular factor that plays a role in the HIV-1 replication cycle. CPSF6 contains an RNA-binding domain that enables its interaction with RNA substrates and other protein components of the CPSF complex, thereby coordinating RNA processing events [13]. CPSF6 shuttles between the nucleus and cytoplasm, displaying dynamic functions in RNA metabolism and gene regulation. In response to viral infections, CPSF6 binds to the HIV-1 CA hexamer of the core to facilitate its nuclear import, which is crucial for infection (Figure 1D and Figure 3) [46,47]. Recent studies have shown that CPSF6 not only impacts the efficiency of viral DNA integration but also influences the selection of integration sites within the host genome [76]. This can affect the stability of the provirus and its susceptibility to transcriptional silencing or activation.

The HIV capsid core is trafficked toward the nucleus (retrograde transport) through the host cell’s microtubule network via the coordinated action of dynein and kinesin motor proteins, which facilitate the directional movement of the capsid core toward the nuclear pore complex (NPC) (Figure 1C) [77,78,79,80]. Microtubule-associated proteins, MAP1A and MAP1S, interact with the CA, playing a role in linking the CA protein to the host cell’s microtubule network [81]. Protein bicaudal D2 (BICD2), a dynein adaptor protein, binds to HIV-1 CA and recruits dynein and dynactin to form a protein complex, facilitating retrograde transport [82,83]. Fasciculation and elongation factor zeta 1 (FEZ1), a kinesin-1 adaptor protein, promotes retrograde transport by bridging kinesin-1 and the CA core through its interaction with the CA hexamer [84,85].

Nup358, located on the cytoplasmic filaments of the NPC, interacts with the HIV-1 CA to facilitate the docking of the viral core at the NPC (Figure 1D). This process is aided by kinesin-1 motor proteins such as KIF5B, which transport the viral core towards the nucleus [86]. Nup358 binds to the CypA binding loop of the HIV-1 CA via its C-terminal cyclophilin domain [87,88]. CPSF6 has been reported to enhance the interaction between Nup358 and the HIV-1 core [86]. Nup153, an essential component of the nuclear basket of the NPC, also interacts with CA hexamers, specifically with a hydrophobic pocket between adjacent CA monomers through its phenylalanine-glycine (FG) motifs (Figure 3) [47,48,89], while a unique triple-arginine (RRR) motif targets the tri-hexamer interface [90]. These interactions are essential for stabilizing the HIV-1 core and facilitating its passage through the NPC [91]. At the NPC, CPSF6 and Nup153 collaborate to facilitate the translocation of the core into the nucleus. CPSF6 assists in releasing the viral core from the NPC following the initial docking mediated by Nup153 (Figure 1D) [92].

Human Myxovirus resistance protein 2 (Mx2/MxB), a dynamin-like large GTPase of the dynamin superfamily, was initially identified as a regulator of cell-cycle progression and cytoplasmic-nuclear transport [93]. MxB also inhibits HIV-1 infection by blocking several stages of the virus replication cycle, most notably nuclear import [91,94,95,96,97,98,99,100], through binding to Nup358, masking its interaction with the HIV-1 core and, thus, impeding the nuclear import of the viral core [101].

### 3.3. Host Metabolites in HIV-1 Maturation and Core Association

HIV-1 can also exploit host cell metabolites for its replication [102,103], including inositol hexakisphosphate (IP6), a cellular metabolite derived from inositol, through a series of phosphorylation reactions, which is involved in both HIV-1 assembly and maturation [104]. Inositol undergoes phosphorylation by various kinases to produce inositol monophosphate (IP1), inositol bisphosphate (IP2), and inositol triphosphate (IP3). Inositol polyphosphate multikinase (IPMK) then catalyzes the conversion of IP3 to inositol tetrakisphosphate (IP4) and IP4 to pentakisphosphate (IP5). Ultimately, IP5 2-Kinase (IPPK) phosphorylates IP5 to produce IP6, which promotes the assembly of both the immature virus particle and the formation of the fullerene core during virus maturation [46,105]. In addition, IP6 promotes the assembly of the immature Gag lattice by binding to two lysine rings in a six-helix bundle that forms at the CA-SP1 junction during particle assembly, thereby driving the incorporation of IP6 into the mature particle [105,106,107,108]. This interaction aids in the cleavage of the Gag polyprotein by the PR and stabilizes the CA structure, thus influencing the maturation of the virus and its ability to infect new cells [109,110,111,112]. During virus maturation, IP6 binds the electropositive pore at the center of the mature CA hexamer and pentamer formed by arginine at position 18 (Arg18) and lysine at position 25 (Lys25) (Figure 3) [112,113]. In the absence of IP6, IP5 also interacts with the Gag polyprotein and is packaged into the budding virion at similar levels as IP6 [108,111,114]. IP5 is coordinated at the center of the HIV-1 hexamer through a ring of Arg18 during maturation (Figure 3) [46,105]. IP5 is located at the center of the six-fold axis, resulting in the averaging of six equivalent binding positions due to symmetry. Unlike IP6, which is less planar because of its axial phosphate, the density of IP5 is much clearer, allowing it to be distinctly positioned in a parallel stacking arrangement with the Arg18 ring (Figure 3). The additional axial phosphate in IP6 would be oriented away from the Arg18 ring, leading to the substitution of IP5 for IP6 without reducing infectivity, which is likely because the interaction is primarily driven by the equatorial phosphates, particularly when the ligand is in a planar conformation [105,108]. FEZ1, as discussed in Section 3.2, competes with IP6 through its interaction with the ARG18 ring of the CA hexamer (ref.). Viral particles produced without sufficient IP6 are noninfectious because CA assembly is inefficient, resulting in deficient core formation [105,115]. Furthermore, IP6 stabilizes the CA core in the target cell, thereby promoting the completion of viral DNA synthesis inside the core [108,116]. Overall, IP6 is an essential element in generating infectious virions by supporting immature particle assembly, Gag and Gag-Pol polyprotein cleavage, and conical core formation [33,115].

### 3.4. Host Membrane Lipid Rafts in HIV-1 Maturation

The composition of the cell membrane plays a crucial role in cellular processes such as endocytosis and the movement of materials into the cell [117]. The plasma membrane contains a variety of lipids distributed asymmetrically across the bilayer [118]. Phosphatidylinositol-4,5-bisphosphate [PI(4,5)P_2_] plays a key role in the recruitment of Gag to the plasma membrane [119]. HIV-1 particles bud from lipid rafts, which are microdomains in the plasma membrane of infected cells rich in cholesterol, sphingolipids, and glycolipids [86]. Therefore, the viral membrane is enriched in cholesterol, sphingomyelins (SMs), glycosphingolipids, and phosphoinositides such as PI(4,5)P_2_ [117]. Neutral sphingomyelinase 2 (nSMase2), an enzyme that generates the bioactive lipid ceramide through the hydrolysis of membrane SMs, plays a critical role in the late stages of HIV-1 maturation and replication [120,121]. nSMase2 regulates the size and stability of membrane microdomains and interacts with HIV-1 Gag to hydrolyze SMs into ceramide, which is necessary for virion maturation [122]. The inhibition of nSMase2 in HIV-1 producer cells results in the production of immature HIV-1 particles with increased SMs and a reduced ceramide content, resulting in defective Gag polyprotein processing and impaired viral envelope formation. This, in turn, severely reduces infectivity [122].

### 3.5. Environmental Factor in HIV-1 Maturation

Environmental factors, such as pH, can contribute to the maturation efficiency of virions. The proteolysis of the Gag polyprotein in virus-like particles (VLPs) by exogenously added recombinant PR has been found to be optimal at pH 6.0 to 7.0 [123], while purified PR is poorly active at neutral pH [124,125]. The internal pH of the HIV-1 virion is generally predicted to be slightly acidic, ranging from pH 6.0 to 7.0 [126]. This pH range ensures the proper processing of the Gag polyprotein as it is optimal for HIV-1 PR activity. Additionally, the cell entry efficiency of enveloped viruses is affected by pH [127], and most enveloped viruses are inactive and unable to invade target cells under acidic conditions of pH 5.0 to 6.0 [128]. Therefore, HIV-1 virion maturation may be controlled through the pH-dependent activation and deactivation of PR.

## 4. Tools for Assessing Virion Maturation

Although there are many assays for assessing HIV-1 maturation, including western blotting, mass spectrometry, flow cytometry, RT- and PR-based assays, and viral infectivity assays, microscopy is the most utilized and quantitative method. In this section, we specifically discuss the approaches for studying HIV-1 maturation using electron microscopy (EM) and fluorescence microscopy.

### 4.1. Electron Microscopy

Microscopy is a critical technique in the analysis of viruses due to its ability to reveal events in the virus replication cycle, even with the relatively small size of virions [129]. Transmission EM (TEM) is a widely used technique due to its ability to determine the number of virions in a sample and characterize their morphology at nanometer-scale resolution by transmitting electrons through a specimen embedded in plastic [130,131,132]. TEM has been applied to HIV research to visualize the morphogenesis and maturation of the virus (Figure 4). TEM can visualize the rearrangement of structural proteins as the virus transforms into a mature, infectious form. Additionally, TEM has been used to evaluate the efficacy of inhibitors that block HIV-1 maturation [32,129,133,134]. TEM contributes to studying the effectiveness of these inhibitors by directly revealing their effect on virion morphology. Insights into the localization of the Gag polyprotein precursor in relation to viral gRNA have also been uncovered by TEM [135]. Cryogenic EM (Cryo-EM) is used to determine high-resolution structures of biological macromolecules in three dimensions (3D) [135]. This technique uses frozen macromolecules fixed in ice to determine their visual projection from different directions. The samples are then processed in two dimensions (2D) before being reconstructed into a 3D form for analysis [135]. Cryo-EM can capture multiple frozen particles in different orientations, providing high-resolution images of individual macromolecular complexes with atomic-level details. On the other hand, cryo-electron tomography (Cryo-ET) takes sequential images of a single particle as it is rotated or tilted along an axis. These series of images are aligned to computationally reconstruct a 3D representation, or tomogram, of the particle.

Cryo-EM/ET has also been used to investigate how viruses operate in the context of viral replication and maturation [136,137,138]. It has also been applied to research surrounding HIV-1 to explore how the virus assembles inside host cells and how different intermediate states form [100]. It has also provided structural insights into the transport of the cone-shaped CA core through the nuclear pore [139] and the variation and positioning of HIV-1 Env on the Gag lattice [140].

### 4.2. Fluorescence Microscopy

Fluorescence microscopy is a key technology adapted to explore the structure of cells, tissues, and other biological components [141]. Super-resolution microscopy techniques surpass the diffraction limit of traditional fluorescence microscopy, enabling the visualization of structures at a nanometer scale, thereby allowing a detailed examination of biological processes at a molecular level [142]. However, even with super-resolution techniques, fluorescence microscopy does not have sufficient resolution to reveal the structural changes that occur in the virion during maturation.

The Förster resonance energy transfer (FRET) principle is a specific feature of fluorescence microscopy, where energy is transferred from an excited fluorophore (the donor) to another fluorophore (the acceptor) [143]. This process is distance-dependent and occurs through an intermolecular long-range dipole–dipole coupling [144]. FRET is an effective technique for investigating various biological processes and has been applied to HIV studies due to its high sensitivity to changes in molecular proximity [134,144,145]. An automated method for detecting PR activity in single virions using fluorescence microscopy was developed [146]. This system employs tandem fluorescent proteins, separated by HIV-1 PR cleavage sequences, fused in-frame to the N-terminus of HIV-1 Vpr. This method utilizes an FRET pair of fluorescent proteins to tag Vpr, providing a convenient reporter for viral PR activity, indicated by a loss of FRET due to the separation of the donor and acceptor proteins. This single-virion FRET-based technology demonstrates that the FRET levels and the relative intensities of the donor and acceptor signals in individual virions can effectively differentiate between immature and mature virions. Although it results in a modest reduction in infectivity, this bifunctional marker may reduce maturation efficiency. Consequently, it serves well for the qualitative differentiation between mature and immature virions but is not suitable for quantitative assessment. Another fluorescence microscopy tool utilizing the FRET principle (HIV-1 Gag-iFRET) was recently used to distinguish between mature and immature virions (Figure 4) [31,134]. This was accomplished by inserting the efficient intramolecular FRET pair combination, ECFPΔC11-cp173Venus [147,148], into the Gag polyprotein between the MA and CA domains, surrounded by an HIV-1 PR cleavage site, based on the HIV-1 Gag-iGFP construct [149] (Figure 2C). In the HIV-1 Gag-iFRET progeny virions, the Gag polyprotein and FRET pair are cleaved by PR during maturation. Separation of the FRET pair results in a reduction in the FRET signal (Figure 4). Alternatively, if maturation is blocked with the treatment of PIs, the FRET pair remains uncleaved and emits an FRET signal [150,151]. This HIV-1 Gag-iFRET virus did not lose any infectivity compared to the unlabeled virus [134] and directly detected the PR-mediated cleavage of Gag and Gag-Pol polyproteins, making it suitable for quantitative analysis without any artificial disruption of maturation. This new fluorescence microscopy-based system showed that approximately 80% of virions completed the cleavage of the MA-CA junction in the Gag and Gag-Pol polyproteins, indicating that they are predicted to be mature virions. This correlated well with the range of virion maturation ratios assessed by TEM results, which vary from 50% to 99%, depending on the laboratory’s manual procedures for determining the morphology of virions [49,55,152,153,154]. Based on a computational analysis of fluorescence emitted at the single-virion level, this FRET-based method enables an unbiased quantification of the proportion of Gag and Gag-Pol polyprotein cleavage by PR inside progeny virions. HEK293T cells, derived from human kidney cells, are typically used for in vitro virus production and the characterization of virion maturation [155]. However, it has recently been discovered using the HIV-1 Gag-iFRET system that the maturation rate of viruses produced from HEK293T cells is relatively higher than in other cells that are more biologically relevant to HIV production [31]. The Gag processing efficiency in viruses from T-cell lines, such as Jurkat cells, was significantly lower than that of viruses from HEK293T cells (68.7% vs. 81.8%, respectively). This difference was also correlated with the virus infectivity [31]. The virus morphological analysis produced by another T-cell line, MT-4 cells, showed that 62% of the virions were mature [112], which aligned with the results from the HIV-1 Gag-iFRET detection system in Jurkat cells. There is limited information about the efficiency of virion maturation in primary T cells, so further characterization and a careful quantitative analysis of maturation efficiency in relevant cell types is warranted. It is also important to note that Gag processing efficiency is an indirect measure of virus maturation efficiency. Therefore, a combination of techniques, including direct morphological analysis by EM and quantitative fluorescence microscopic assays, will comprehensively address this important matter.

## 5. Influence of Virion Maturation on HIV-1 Env Function

The HIV-1 Env mediates the fusion between viral and cellular membranes (Figure 1A,B). Upon the binding of gp120 to CD4, conformational changes occur in the Env trimer, exposing the binding sites for co-receptors (CCR5 or CXCR4) on gp120. The subsequent interaction of gp120 with the co-receptor facilitates the insertion of the fusion peptide of gp41 into the target cellular membrane. The fusion between viral and cellular membranes is mediated by the formation of an antiparallel 6-helix bundle involving heptad repeat 1 and 2 (HR1 and HR2) in the ectodomain of gp41 [156]. Single-molecule FRET (smFRET) analysis has revealed that Env exists as metastable trimers on virions and infected cells, sampling at least three intramolecular conformations: state 1 (pre-triggered, closed), state 2 (functional, intermediate), and state 3 (fully CD4-bound, open) [157,158,159,160].

Env is incorporated into virions as a trimer of gp120-gp41 heterodimers [161]. The MA domain of Gag plays a pivotal role in Env incorporation during viral assembly [10]. The MA domain forms a trimeric arrangement [162] and multimerizes into a hexameric lattice of trimers on PI(4.5)P_2_-containing membranes, as well as within immature and mature virions [42,163]. Several studies indicate that MA trimerization is critical for efficient Env incorporation [164,165]. Although direct structural evidence for the interaction is still lacking, biochemical studies suggest that the long gp41 cytoplasmic tail (CT), which is approximately 150 amino acids in length, interacts with the central gap of the MA lattice [164,165]. Env in immature virions is fusion-incompetent, and the PR-mediated Gag cleavage at the SP1-NC junction, which releases the downstream SP1 from MA, is required to activate fusogenicity [166,167]. The suppression of fusogenicity in immature virions can also be rescued by the deletion of the gp41 CT [166,167], suggesting that interactions between the gp41 CT and MA domain of uncleaved Gag inhibit premature fusion. Fluorescence nanoscopic analyses have shown that Gag cleavage by PR induces the redistribution of Env trimers (Figure 1K and Figure 4). Env trimers are dispersed on immature virions but are clustered on mature virions, a process that is dependent on the gp41 CT [168]. Recent cryo-EM analysis has revealed that gp120 binding to CD4 further clusters Env-CD4 complexes in the membrane–membrane interface, with the clustering influenced by viral maturation status [169]. Despite their dispersion, Env trimers on immature virions can adopt an open conformation upon CD4 binding, similar to those on mature virions [170], suggesting that maturation affects intermolecular Env interactions rather than intramolecular Env conformations. However, further analysis is required to understand the impacts of Gag maturation on the Env conformation. Interestingly, immature virions exhibit greater rigidity than mature virions, and virion stiffness is also regulated by the gp41 CT and is correlated with viral infectivity [171]. The composition of the viral membrane also influences the distribution and function of Env trimers. A recent study demonstrates that the gp41 CT directly binds to viral envelope cholesterol [172]. This interaction is required for the Env clustering on the viral membrane and subsequent fusion events [172,173,174]. Several studies have reported that HIV-1 virions typically contain, on average, eight to ten Env trimers, which is a lower density compared to other enveloped viruses such as measles and influenza viruses [175,176]. The low density of Env complexes on HIV-1 particles may be a strategy for evading immune detection. It has been predicted that multiple Env-receptor interactions are required for viral fusion (Figure 1A,B) [177], underscoring the importance of Env clustering for viral fusion. As described above, HIV-1 MA forms hexamers consisting of six trimer units in both immature and mature virions. However, the arrangement of the MA lattice differs between these states, suggesting that Gag cleavage by PR triggers conformational rearrangements of the MA lattice [42]. These findings suggest that structural rearrangements of the MA lattice may influence virion stiffness and the distribution of Env trimers, which are crucial for facilitating effective fusion events.

## 6. Targeting Maturation for Therapeutic Intervention: Small Molecule Discovery, Mode of Action, and Clinical Development

The rationale of the pharmaceutical targeting of virion maturation has been explored for over three decades, with multiple successful approaches leading to the identification of different classes of small molecule inhibitors. Some of these classes, such as PIs, were initially marketed during the second decade of the HIV pandemic, while others, including CAIs, maturation inhibitors (MIs), and allosteric integrase inhibitors (ALLINIs), have been developed in recent years. CAIs exhibit the highest antiviral potency and the longest-acting therapeutic profiles achieved to date.

### 6.1. Protease Inhibitors

The first class of small molecule inhibitors of virus maturation was developed early in the HIV pandemic, with a mechanism that directly blocks the catalytic activity of PR. PIs played a critical role in the development of antiretroviral therapies and were included in three-drug regimens, typically with two NRTIs as the backbone. Soon after the discovery of the PR structure and its function in virus maturation and infectivity, multiple active-site PIs were developed as antiretrovirals, with saquinavir (SQV) being the first approved member of this class in 1995 (Table 1) [178]. A seminal work demonstrated that a single substitution of the aspartic acid residue at position 25 with asparagine (D25N) in the PR catalytic site resulted in the production of immature virions that contained only uncleaved Gag polyprotein while preserving PR expression [179]. These results, along with the availability of crystal structures of PR, demonstrated that the catalytic site of PR was a viable target for the development of inhibitors [180,181]. Seven amino-acid peptides corresponding to specific sequences around Gag cleavage sites are sufficient for the proteolytic activity (Figure 2), suggesting that small peptidic substrates could be investigated as PI candidates [182,183]. Most of the approved PIs were then conceived to mimic the PR substrate and were, for this reason, named peptidomimetics. These inhibitors contain a hydroxyl group that interacts with the carboxyl group of residues in the active site of PR. The inhibition of the catalytic activity of PR blocks Gag and Gag-Pol polyprotein precursor processing, resulting in the formation of immature virus particles that are not infectious. Long-term treatment with first-generation PIs revealed limitations due to the high pill burden, frequent adverse events, and potential for drug–drug interactions. These interactions arise because PIs are substrates for the Cytochrome P450 3A4 (CYP3A4) enzyme, which metabolizes many other drugs in the liver, and, thus, PIs are co-administered with pharmacokinetic boosters such as ritonavir (RTV) and cobicistat (COBI) [184]. RTV, initially approved as a PI, is now used in lower doses primarily for its potent CYP3A4 inhibition [185]. COBI, a structural analog of RTV, lacks antiviral activity but also serves as a strong CYP3A4 inhibitor [186]. Both RTV and COBI increase the plasma concentrations of other antiretrovirals, including PIs, thereby enhancing their efficacy, reducing effective doses, and decreasing pill burden. On the other hand, inhibiting CYP3A4 can lead to increased drug levels in the blood, raising the risk of toxicity and adverse drug interactions by slowing the metabolism and clearance of many medications. Second-generation PIs, such as darunavir (TMC114), which was the last PI approved in 2006 [187] (Table 1), were developed to address some of these issues, offering a greatly improved virologic profile with high genetic barriers to resistance and better tolerance, even though they remain a substrate for CYP3A4. Taken together, while PIs were once central to HIV therapy, they are now less commonly used as first-line regimens due to the side effects, complex dosing requirements, and availability of newer antiretroviral classes with improved safety and efficacy profiles. Continued research is essential for developing next-generation PIs with safer drug profiles [188,189].

### 6.2. Maturation Inhibitors

MIs block particle maturation by binding at the CA-SP1 junction of Gag, thereby preventing CA-SP1 cleavage, which, as mentioned above, is the last cleavage event in the Gag processing cascade and is critical for liberating mature CA (Figure 2). Thus, this class of inhibitors does not block the intrinsic enzymatic activity of PR but rather prevents the enzyme from accessing its substrate. A screening of natural products derived from leaf extracts of *Syzygium claviflorum*, a tree in the Myrtaceae family native to Australia and Asia, found that a known triterpenoid, betulinic acid, and its derivatives had antiviral activity against HIV-1 in assays conducted with H9 lymphocytic cells. Although with a half maximal effective concentration (EC_50_) in the single-digit micromolar range and a modest therapeutic index, the activity of those compounds seemed to be specific against HIV-1 replication [190]. Subsequent medicinal chemistry efforts led to the discovery of the dimethyl succinyl derivative of betulinic acid, known variously as YK-FH312, PA-457, DMB, and now bevirimat (BVM) [191,192,193] (Table 1). The time of addition assays demonstrated that BVM inhibits the late stages of virus replication [191]. A seminal study established that the mode of action of BVM is via inhibiting the conversion of the capsid precursor (CA-SP1 or p25) to the mature form (CA or p24), which induces the formation of virus particles that have an aberrant morphology and are not infectious [193]. Importantly, an early report showed that in vitro resistance selection with BVM identified a single mutation at the N-terminus of SP1, which flanks the CA-SP1 cleavage site [136,194,195,196,197], suggesting that the compound binds at the CA-SP1 junction. Further resistance selections and studies using single amino acid deletions spanning the entire CA-SP1 boundary domain expanded our understanding of the BVM binding pocket, and other critical residues were identified in the C-terminus of CA [198,199,200,201]. By using photoaffinity-labeled analogs of BVM, it was shown that this series targets the Gag major homology region (MHR), which is critical for virus assembly and mediates the binding of Gag to IP6 that stabilizes the CA-SP1 lattice [202]. Overall, these studies confirmed that the target site for BVM resides in the Gag CA-SP1 region. With regards to the antiviral spectrum, BVM was found to be active against a panel of wild-type (WT) and drug-resistant isolates [193] but cannot inhibit simian immunodeficiency virus (SIV) [201,203]. Biochemical assays using recombinant monomeric Gag protein failed to recapitulate BVM activity, suggesting that the BVM binding site is present only in multimeric assembled Gag proteins [193]. Furthermore, it was shown by liquid chromatography–mass spectrometry that the compound can bind immature particles but not bind either immature virions containing BVM resistance mutations or mature virions [204]. Recent insights into BVM-mediated antiviral activity revealed its interaction with a six-helix bundle that forms at the CA-SP1 junction in the immature Gag lattice during particle assembly. The current experimental model is based on a dynamic equilibrium between a stable and a disordered state [32]. Structural information on BVM indicates that the molecule binds within the six-helix bundle, thereby stabilizing it. Since the CA-SP1 cleavage site is buried within the bundle, its stabilization by BVM prevents PR from accessing the CA-SP1 cleavage site [106,136,193,194,196,197,205,206]. As mentioned above, IP6, which also binds and stabilizes the six-helix bundle, is an integral component of proper virus particle maturation. The IP6 binding site in the immature Gag lattice contains two rings of lysine side chains sitting in the six-helix bundle above the BVM binding site. BVM and IP6 have been shown to simultaneously bind to the six-helix bundle [113], restricting the mobility of residues in the CA-SP1 junction. This suggests a cooperative mechanism that restricts PR access to the CA-SP1 cleavage site [205]. Several reports investigating the interdependence between IP6 and MIs suggest an intriguing hypothesis that inhibitors designed to compete with IP6 binding to the bundle may enhance antiviral activity [113].

Five key MIs have undergone evaluation in Phase 1-2b clinical trials. BVM showed a decline in HIV-1 RNA of nearly 0.45 log_10_ c/mL in 10 days in Phase 2a [207]. However, its potency was compromised by several polymorphisms in the CA-SP1 junction region. Another MI, GSK2838232 (Table 1), demonstrated a more potent antiviral decline of about 1.7 log_10_ c/mL [208]. On the other hand, its PK properties required boosting with RTV, which led to the termination of its development. The MI GSK3532795 (Table 1) showed a potent antiviral response regardless of any polymorphism and did not require boosting with RTV [209]. Thus, GSK3532795 has been evaluated in two Phase 2b studies; the dose range-finding study showed a Week 24 efficacy rate of approximately 82% on a backbone of two NRTIs in treatment-naïve adults with HIV-1 [210]. This compound was terminated for high rates of resistance, both to the NRTI backbone and MI itself, and high rates of gastrointestinal (GI) intolerability (e.g., abdominal pain and diarrhea). The MI GSK3640254 (Table 1) was studied through two Phase 2b studies in treatment-naïve adults with HIV-1, one in combination with two NRTIs and another solely administered with an INSTI, dolutegravir (DTG). Notably, both trials showed similar clinical profiles to the reference regimen: similar rates of virologic responders at Week 24 (without any treatment-emergent resistance in either trial) and acceptable safety/tolerability profiles [211]. Ultimately, it was determined that an intended Phase 3 fixed-dose combination of GSK3640254 and DTG would not be superior to the existing two-drug daily oral regimens. Thus, GSK3640254 was not advanced into Phase 3. At the 2024 Conference on Retroviruses and Opportunistic Infections (CROI), an additional MI, VH3739937 (VH-937, also known as GSK3739937), was reported to have antiviral properties against HIV-1 strains with diverse Gag sequences and polymorphisms generally conferring resistance to prior MIs (CROI 2024, Abstract 633). In a Phase 1 study, VH-937 administered to healthy participants was shown to be generally well-tolerated [212].

### 6.3. Other Classes

Although not specifically designed for either the direct or indirect inhibition of PR, two other classes of antivirals have been shown to induce defects in the assembly and function of the viral core, resulting in severely compromised particle infectivity.

#### 6.3.1. Capsid Inhibitors

Members of this class of small molecules have a multimodal activity and block both early and late stages of the virus replication cycle (Figure 1D,H,K). The inhibition that CAIs exert in target cells prior to viral DNA integration predominates at clinically relevant inhibitor concentrations [49,213]. Recent work reported that LEN, the first FDA-approved CAI in this class, increases the rigidity or “brittleness” of the viral core, causing hindrance to its nuclear transport and disassembly, eventually preventing the proper integration of the viral DNA in the host genome (Figure 1D) [214,215]. Although with reduced potency, CAIs are also active in virus-producer cells with mechanisms that are less understood. In this regard, it has been reported that CAIs accelerate the rate of CA assembly in vitro and increase the formation of aberrant, noninfectious particles with spherical, non-conical cores (Figure 1K and Figure 3) [49,213]. Several molecules targeting CA have been developed in the past two decades and can be classified based on their binding site [216]. The most potent compounds are those targeting the hydrophobic pocket that lies between CA monomers that are assembled into a hexamer. This site comprises the α-3 and α-4 helices of the CA_NTD_ and the α-8 and α-9 helices of the CA_CTD_ in the neighboring monomer. Importantly, this pocket mediates the interactions of CA with FG-containing motifs of host factors, including Nup153 and CPSF6 (Figure 3), which perform critical roles in HIV nuclear trafficking and integration (Section 3.2) [47,217,218]. Members of the class of CAIs that bind the FG site include the prototype PF3450074 (PF-74), BI-1, and BI-2, and the more recently developed GSK878, GS-CA1, and LEN (GS-6207) (Table 1). GSK878, GS-CA1, and LEN display remarkable sub-nanomolar half-maximal inhibitory concentrations (IC_50_) [49,213,219,220,221,222]. Resistance to LEN emerges rapidly in vitro and, in most cases, selects for substitutions that reduce virus fitness, with the exception of the low-level resistance-associated variant Q67H [49].

Because of its potent antiviral activity and favorable drug-like properties, LEN was tested in clinical trials with long-acting formulations and was generally well-tolerated. In an early monotherapy Phase 1b study, single subcutaneous (SC) doses of LEN displayed potent antiviral activity over 9 days. A genetic mixture of WT CA and CA-Q67H mutant emerged in some study participants treated with a low dose (20 mg), which was associated with a decrease in phenotypic susceptibility but without any viral escape noted by day 9 [49]. Subsequently, LEN has also been tested as a twice-yearly SC injectable treatment in combination with other once-daily oral antiretrovirals in a Phase 2 study in treatment-naïve PLWH (CALIBRATE; NCT04143594). This trial confirmed LEN’s safety and potency and showed that emergent resistance, including the Q67H variant, was rare and attributed to an incomplete adherence to the daily oral agents [223]. The Phase 2/3 CAPELLA study (NCT04150068) is being conducted in heavily treatment-experienced (HTE) PLWH with multi-drug resistance. The trial tested an initial 14-day oral administration of LEN alongside the failing therapy, followed by twice-yearly SC injections of LEN with an optimized background regimen (OBR) during the maintenance period starting on day 15. Interim data confirmed LEN’s safety and potency, showing that resistance emerged when LEN was unintentionally used as functional monotherapy due to either insufficient adherence to the OBR or the absence of fully active antivirals in the OBR [224,225,226]. Although LEN, like other CAIs, may have a lower genetic resistance barrier than other classes of antiretrovirals, such as INSTIs, its potency is the best among all antiretrovirals approved to date, and its drug-like properties are favorable for long-acting oral and injectable formulations. For these reasons, in 2022, LEN received approval from the FDA for clinical use in HTE adults who have limited treatment options, and it is being offered as a twice-yearly injection administered in addition to an OBR of once-daily oral antiretrovirals. However, LEN offers the potential for a long-acting regimen if a suitable partner molecule can be developed. Currently, the longest-duration regimen approved for treatment is a once-every-two-month intramuscular (IM) injection of Cabenuva [227,228], although clinical studies are underway to extend the duration to four-month intervals. In addition, LEN is also being investigated for PrEP, and the PURPOSE 1 Phase 3 clinical trial using a twice-yearly SC administration of LEN demonstrated remarkable efficacy and superiority to a daily oral regimen [229]. Currently, the only long-acting agent approved for PrEP is Apretude, an extended-release injectable suspension of CAB that is administered intramuscularly once every two months [230,231].

#### 6.3.2. Allosteric Integrase Inhibitors

The IN is an enzyme with endonuclease and polynucleotide transferase activities present in the catalytic core domain (CCD) that is located between the N-terminal and C-terminal domains of the mature protein. Two sequential catalytic reactions are carried out by retroviral INs: the first processes the 3′ ends of the newly synthesized double-stranded viral DNA, exposing a dinucleotide sequence, and the second is the DNA strand transfer reaction. The latter mediates a covalent ligation of the processed 3′ ends into the host genome [232,233,234,235]. Several INSTIs have become critical components of modern ART due to their high genetic barrier to resistance and several favorable drug-like properties, including long-acting potential [236]. Relevant to this review is a more recently identified secondary activity that IN exerts late in the virus replication cycle by binding to the viral gRNA and promoting its localization within the viral core [237,238]. Early mutagenesis studies identified residues within the CCD that are critical for the enzymatic activity of IN and that are known as class I mutations. Additionally, other sequences outside of IN’s active site were found, whose mutations pleiotropically inhibited the late stages of virus assembly. These were classified as class II mutations [239]. In virus-producer cells, these substitutions induced the formation of virions in which the viral gRNA failed to be packaged inside the core but rather was present as an electron-dense nucleoid located outside of the fullerene cone [239,240]. In the early 2000s, a host factor, lens epithelium-derived growth factor/p75 (LEDGF/p75), was discovered to bind IN, favoring integration in transcriptionally active regions of the host chromatin [241,242,243,244]. Screening for inhibitors of the LEDGF-IN interaction led to the discovery of compounds that were originally called LEDGF-IN inhibitors (LEDGINs) [245,246,247,248,249]. Upon further study, it became clear that the primary mechanism of action of these compounds was not to inhibit integration but rather to disrupt the IN-mediated packaging of the viral gRNA in the core during maturation. It was shown that ALLINIs induce the hyper-multimerization of IN, resulting in virions whose aberrant morphology resembled that of class II IN mutations mentioned above [250,251,252]. These compounds are now predominantly referred to as ALLINIs. X-ray crystallographic analysis demonstrated that ALLINIs mediate the interface between head-to-tail interactions of CCDs in one IN dimer and the C-terminal domain of a second dimer, resulting in the defective multimerization of the protein [250,253,254,255]. The pharmaceutical development of ALLINIs recently led to pirmitegravir (Table 1), also referred to as STP0404, the first molecule in this class advanced to the clinic, an investigational compound currently in Phase 2 clinical development [256].

## 7. Conclusions

Several different classes of inhibitors have been discovered to target virion maturation and prevent HIV-1 replication. These include PIs, RTIs, INSTIs, MIs, CAIs, and ALLINIs (Figure 1 and Table 1) [257]. The current standard of HIV treatment follows a daily therapy regimen, which could negatively impact those who may not be able to adhere to the daily regimen or neglect to take medications [258,259]. In 2021, the FDA approved the first long-lasting ART regimen, Cabenuva, which consists of CAB and RPV [7]. This regimen requires IM injections of co-packaged CAB and RPV, administered either once a month or every two months. In the further ACTG A5359 LATITUDE Phase 3 trial, a once-monthly IM administration of CAB/RPV demonstrated superior efficacy compared to the daily oral standard of care in PLWH with a history of adherence challenges (NCT03635788; CROI 2024, Abstract 212). Additionally, the IMPAACT 2017 study (MOCHA; NCT03497676) showed that the use of long-acting CAB/RPV was generally well-tolerated in adolescents with HIV (CROI 2024, Abstract 188). In 2022, the FDA approved the first-in-class CAI, LEN, which involves a twice-yearly dosing in combination with a daily OBR for the treatment of HTE PLWH. LEN has a multi-stage mechanism of action, including targeting the late stage of virus replication and interfering with virion assembly and maturation (Figure 1H,K). LEN is specifically effective for the treatment of multidrug-resistant HIV-1 strains. If combined with a suitable long-acting partner, which is yet to be found, LEN has the potential to be included in long-acting regimens for HIV treatment against multidrug-resistant strains. In addition, LEN is currently under investigation for PrEP [260], which could potentially contribute significantly to the prevention of new infections, ultimately working towards the elimination of HIV from humanity. Overall, targeting virion maturation is important for developing long-acting antivirals, especially against multi-drug-resistant mutants, and for providing new options for the treatment and prevention of HIV infection. Currently, there are no FDA-approved MIs for HIV treatment, but VH-937 has shown potential to become the first approved MI in ongoing clinical trials. Further studies on this new compound are highly anticipated.

### Future Prospects

A better understanding of virion maturation is critical for the development of antivirals. For this purpose, the establishment of non-biased experimental tools is required to systematically investigate HIV-1 maturation. Though TEM is the most direct experimental technique used to address this question, the rates of virion maturation have been reported in a varied range and have not been clearly determined [49,55,152,153,154]. As described in Section 4, an unbiased virion visualization system utilizing the FRET principle to quantitatively detect the cleavage of Gag and Gag-Pol polyproteins by PR, particularly at the MA-CA junction, has been established [134]. This system serves as an indicator to distinguish between mature and immature HIV-1 virions. (Figure 4). However, the current system visualizing HIV-1 Gag-iFRET labeling viruses is limited, as it can only detect the initial step of maturation for Gag and Gag-Pol polyprotein cleavage by PR. To overcome this limitation, a second-generation HIV-1 Gag-iFRET system that can detect proper conical core formation is under development. When the GFP fluorescent protein is fused with CA and is not cleaved by PR during maturation, the GFP-CA fusion protein is incorporated in the conical core (HIV-1 Gag-iHGFP) [55]. Additionally, the free GFP-CA interacts with viral gRNA that is packaged in the closed core [55]. Based on these features, it is hypothesized that engineering cp173Venus in the HIV-1 Gag-iFRET construct, in fusion with CA and lacking the PR cleavage site, can label closed conical cores after Gag polyprotein cleavage. This new construct expands on the HIV-1 Gag-iFRET system by its ability to detect both (1) the processing of the Gag and Gag-Pol polyprotein and (2) the proper CA assembly to form a core during HIV-1 virion maturation [32]. Through the establishment of this new system, a non-biased, semi-automatic, and quantitative assessment of the full maturation process can be achieved in combination with EM analysis. An FRET tool based on single virion analysis can be applied to better understand the mechanism of action of CAIs and MIs currently under development and to further investigate next-generation antivirals targeting virion maturation. This approach aids in the development of long-acting anti-HIV drugs that can improve the quality of life for PLWH.

## Figures and Tables

**Figure 1 viruses-16-01423-f001:**
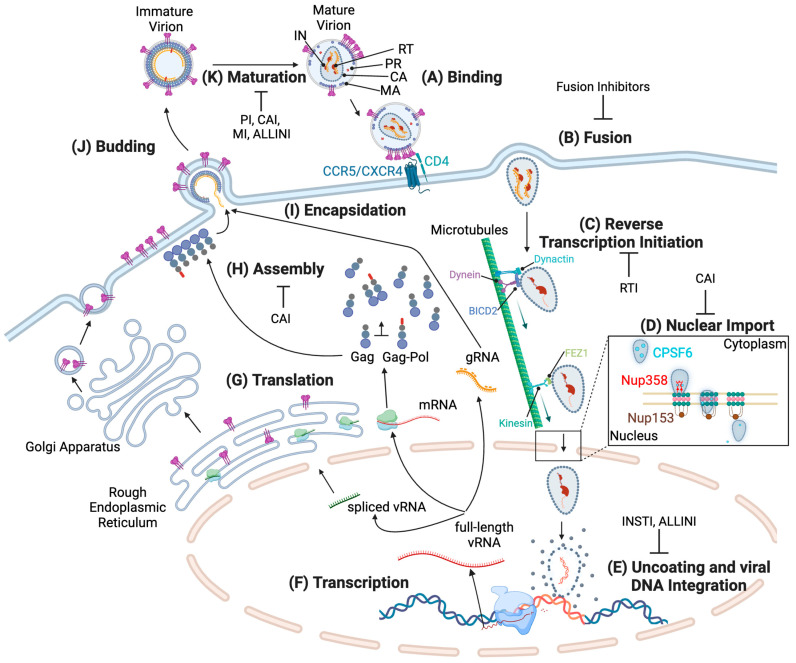
HIV Replication Cycle. Schematic representation of the HIV replication cycle. (**A**) HIV-1 Env binds to plasma membrane receptors, CD4, and CCR5 or CXCR4 coreceptors. (**B**) Upon interactions between Env and host receptors, viral and host membrane fusion occurs, releasing the viral core into the host cell cytoplasm. (**C**) Reverse transcription is initiated shortly after virus entry. The HIV-1 CA core is transported toward the nucleus through the host cell’s microtubule network by dynein and kinesin motor proteins. (**D**) Reverse-transcribed intermediate products inside the intact core are imported into the nucleus through the nuclear pore. (**E**) Upon completion of reverse transcription, viral DNA induces uncoating of the core structure in the nucleus, and IN mediates viral DNA integration into the host genome. (**F**) Proviral DNA is transcribed by the host machinery, enhanced by HIV-1 Tat. (**G**) Viral mRNAs are translated by free ribosomes or ER-associated ribosomes specific for spliced forms encoding the env gene. (**H**) Gag and Gag-Pol polyproteins assemble at the plasma membrane and recruit Env in progeny virions. (**I**) Full-length viral RNAs migrate to the plasma membrane and are packaged into progeny virions. (**J**) Progeny virions bud from virus-producer cells. (**K**) PR cleaves Gag and Gag-Pol in immature virions, releasing CA proteins that form the conical core harboring viral gRNA. Viral Env distribution changes during maturation. The currently developed antivirals disrupt multiple steps of the viral replication cycle, including (**B**) Fusion, (**C**) Reverse Transcription, (**D**) Nuclear Transport/Import, (**E**) Integration, (**H**) Assembly, and (**K**) Maturation.

**Figure 2 viruses-16-01423-f002:**
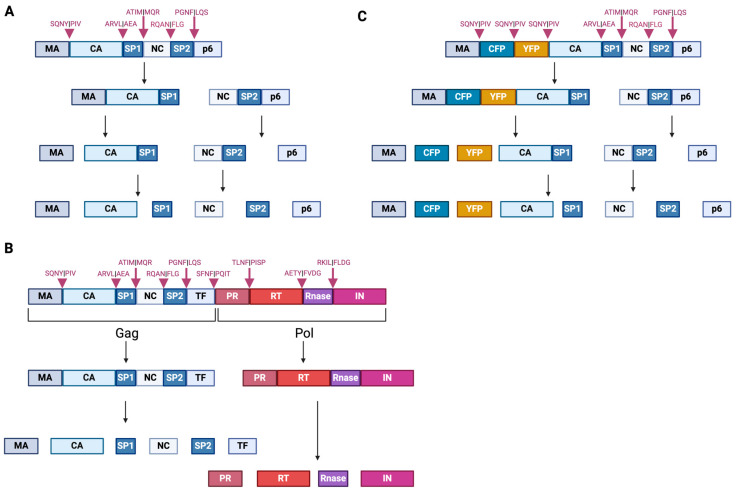
Gag and Gag-Pol polyprotein processing by PR. Schematic representation of (**A**) Gag and (**B**) Gag-Pol polyprotein cleavage pattern by PR. (**A**) The first cleavage occurs between the SP1-NC region within the RQA|NFLG protease recognition sequence, followed by cleavages between MA-CA within SQNY|PIV and SP2-p6 within PGNF|LQS. Finally, SP1 and SP2 short peptides are removed from CA and NC, respectively, within the ARVL|AEA and RQAN|FLG protease recognition sequences. (**B**) The Gag-Pol polyprotein is formed by a frameshifting event at the NC/SP1 boundary, which translates the p6 domain in an alternative reading frame (Transframe/TF). The initial cleavage events separate Gag and Pol regions by cleaving between TF and PR within the SFNF|PQIT protease recognition sequence, carried out by the poorly active PR precursor. This process frees the protease from the rest of the precursor, enabling the ordered processing of the Gag polyprotein. (**C**) FRET indicator fluorescent proteins inserted between the MA and CA domains, bridged by the SQNY|PIV protease cleavage sequence. CFP and YFP separate when MA-CA is cleaved by PR.

**Figure 3 viruses-16-01423-f003:**
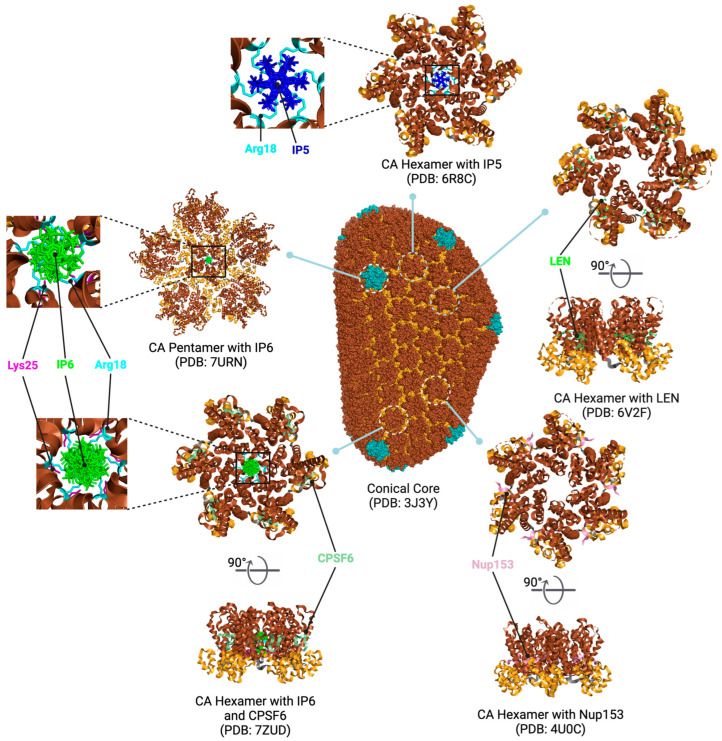
Structural representation of the HIV-1 CA Core. Approximately 250 hexamers and exactly 12 pentamers of CA assemble into a conical core (PDB: 3J3Y) [43,44]. The N-terminal domain from residues 1–145, the linker between residues 146–150, and the C-terminal domain from residues 151–231 of CA are colored in brown, gray, and bright orange, respectively. Twelve pentamers are specifically highlighted in sky blue. IP6 (green) is incorporated into both capsid hexamers (PDB: 7ZUD) [44,45] and pentamers (PDB: 7URN) [45]. In the absence of IP6, IP5 (blue) is incorporated into the capsid hexamers (PDB: 6R8C) [46]. NUP153 (pink in PDB: 4U0C) [47,48]), CPSF6 (pale green in PDB: 7ZUD), and LEN (green in PDB: 6V2F) [49] associated with the capsid hexamer are highlighted. To dynamically visualize these host proteins and small molecule interactions with the CA hexamer, the angles are rotated by 90 degrees from the outer surface angle on the x-axis.

**Figure 4 viruses-16-01423-f004:**
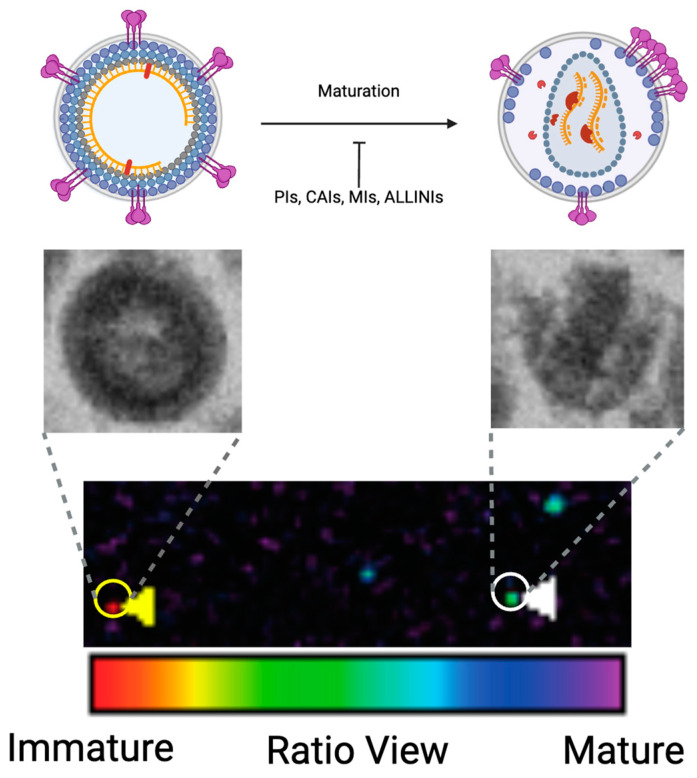
Schematic, TEM, and fluorescence images of HIV-1 Gag-iFRET viruses. While immature virions, depicted by both schematic and TEM images, exhibit a higher FRET signal (red in the ratio color; yellow arrow), mature virions show a low FRET signal (blue in the ratio color; white arrow). Under electron microscopy, HIV-1 immature virions are characterized by an incomplete, spherical structure, while mature virions display a conical core indicative of successful proteolytic processing and proper assembly of the capsid proteins. Under fluorescence microscopy, immature virions express a red signal, which is reduced in mature virions in ratio view. This results in mature virions emitting a blue and green signal. Several drugs, including PIs, MIs, CAIs, and ALLINIs, interrupt virion maturation.

**Table 1 viruses-16-01423-t001:** Antivirals targeting virion maturation.

Class	Structure	Stage of Development
Protease Inhibitors (PIs):
Saquinavir (SQV)	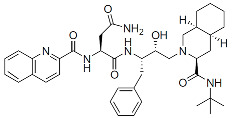	FDA approved, 1995
Darunavir (DRV)	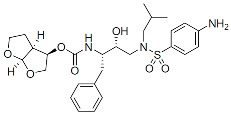	FDA approved, 2006
Maturation Inhibitors (MIs):		
Bevirimat (BVM)	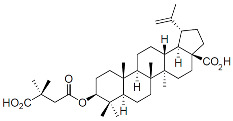	Clinical, terminated
GSK2838232	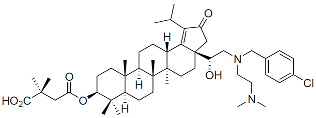	Clinical, terminated
GSK3532795	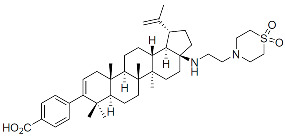	Clinical, terminated
GSK3640254	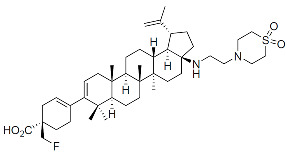	Clinical, terminated
Capsid Inhibitors (CAIs):
PF-3450074 (PF74)	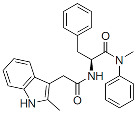	Discovery
BI-1	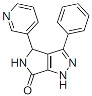	Discovery
BI-2	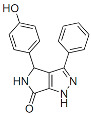	Discovery
GSK878	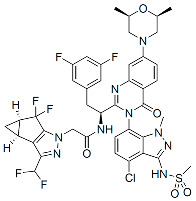	Pre-clinical
GS-CA1	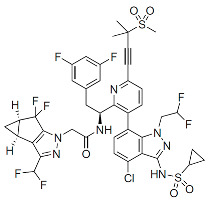	Pre-clinical
Lenacapavir (LEN)	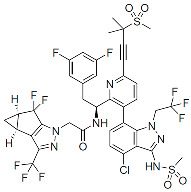	FDA approved, 2022
Allosteric Integrase Inhibitors (ALLINIs):
Pirmitegravir (PIR)	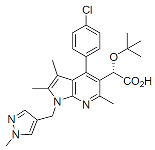	Clinical, Phase 2

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
