# Peer review of "Exploring HIV-1 Maturation: A New Frontier in Antiviral Development"

_viruses, 2024, doi:10.3390/v16091423_

Round 1

Reviewer 1 Report

Comments and Suggestions for Authors

The manuscript by McGraw et al. summarizes the current understanding of HIV-1 maturation process and the intermediate steps as potential targets for discovering new HIV-1 drugs. This review is timely and well-written, and it covers the virus' structural proteins that are not the primary targets of widely used HIV-1 drugs, which target the viral enzymes. Chapter 4 discusses the key techniques of FRET and TEM that have been critical in revealing HIV-1 maturation steps. Chapter 5 discusses the roles of trimeric Env protein. Chapter 6.2 highlights the maturation inhibitors that primarily restrict PR access to CA-SP1 cleavage site. Chapter 6.3 focuses on capsid inhibitors that alter the assembly and disassembly characteristics of capsid core structure.

Suggestions:

1.        The authors may consider including the processing of Gag-Pol in Figure 2.

2.        Figures or cartoons representing the mode of action of a maturation inhibitor (Chapter 6.2) and a Capsid inhibitor (Chapter 6.3) can be valuable.

Author Response

Thank you for reviewing our manuscript and providing positive comments along with a high evaluation. We sincerely appreciate your time and feedback.

Below, we have outlined our responses to your comments.

Comment 1: The authors may consider including the processing of Gag-Pol in Figure 2.

Response to Comment 1: We have added Figure 2B to illustrate the Gag-Pol processing, and the original Figure 2B has been moved to Figure 2C. Additionally, we have included a description of the Gag-Pol polyprotein processing in the text on lines 220-226.

Comment 2: Figures or illustrations representing the mode of action of a maturation inhibitor (Chapter 6.2) and a Capsid inhibitor (Chapter 6.3) could be valuable.

Response to Comment 2: We have created a graphical abstract (GA) that outlines the mode of action for each inhibitor, including MIs and CAIs, which affect virion maturation. We believe this GA addresses the reviewer’s suggestion.

Reviewer 2 Report

Comments and Suggestions for Authors

Good overview covering HIV maturation-, protease-, capsid- inhibitors and ALLINIs.  Nice summary of the HIV maturation process and host factors that contribute to HIV maturation. I applaud the authors for including a section on tools for evaluating virion maturation.  

Some very minor revisions:

Author superscripts:

-'4' is not properly defined- it appears as a '2' instead of a '4'.

-star indicating authour correspondence- this star appears alongside two authors names whilst only a single email is provided.

Author Response

Thank you for reviewing our manuscript and providing positive comments along with a high evaluation. We sincerely appreciate your time and feedback. Below, we have outlined our responses to your comments.

Comment 1: -'4' is not properly defined—it appears as a '2' instead of a '4'.

Response to Comment 1: Thank you for identifying this typo. We have corrected it accordingly.

Comment 2: -Star indicating author correspondence—this star appears alongside two authors' names while only a single email is provided.

Response to Comment 2: Thank you for the suggestion. We have added another corresponding author's email address in the revised manuscript.

Reviewer 3 Report

Comments and Suggestions for Authors

The review article, “Exploring HIV-1 Maturation: A New Frontier in Antiviral Development,” by Aidan McGraw and colleagues, examines the critical role of virion maturation in the viral replication cycle. Additionally, it summarizes how host factors influence the HIV maturation process and provides an overview of various classes of inhibitors targeting virion maturation, detailing their mechanisms of action, the current status of their clinical development, and emerging technologies aimed at deepening our understanding of these mechanisms.

I enjoyed reading this review article and believe it will be a valuable addition to the existing literature. I have only two minor suggestions for edits, though the article is already strong enough to be published as it is.

  1. Perhaps it is just the version provided, but some images/tables appear blurry in certain areas (e.g., Table 1 and the lower part of Figure 4).

  2. In Section 3.2 (lines 193-254), the authors could mention that "in addition to the factors discussed, there are other host proteins involved in maturation and core association that will not be further addressed." I believe their selection of factors to focus on is appropriate and reasonable; this addition would simply ensure the reader does not perceive the list as exhaustive.

Author Response

Thank you for reviewing our manuscript and providing positive comments along with a high evaluation. We sincerely appreciate your time and feedback. Below, we have outlined our responses to your comments.

Comment 1: Perhaps it is just the version provided, but some images/tables appear blurry in certain areas (e.g., Table 1 and the lower part of Figure 4).

Response to Comment 1: Thank you for bringing this to our attention. The resolution of all figures and tables on our end is very clear, and we exported all images at 300 dpi. However, we have independently re-uploaded each figure and table in this revision to ensure optimal clarity.

Comment 2: In Section 3.2 (lines 193-254), the authors could mention that "in addition to the factors discussed, there are other host proteins involved in maturation and core association that will not be further addressed." I believe their selection of factors to focus on is appropriate and reasonable; this addition would simply ensure the reader does not perceive the list as exhaustive.

Response to Comment 2: Thank you for the suggestion. As noted, we believe we have described the majority of HIV-1 core-associated host factors that influence virus maturation and subsequent infection in this section. Following your suggestion, rather than adding the specific sentence proposed, we have included a discussion on the CA core association with microtubules for retrograde transport (lines 285-294). We believe this addition provides adequate coverage of the most relevant CA core-associated host factors. 

Reviewer 4 Report

Comments and Suggestions for Authors

McGraw et al. wrote an excellent review on HIV-1 maturation. This article nicely summarizes current knowledge on late stages of the HIV replication cycle, in particular HIV maturation, and drugs available to prevent these steps of the HIV life cycle. This review is comprehensive and thoroughly covers the field and would help readers better understand HIV maturation. The reviewer has only minor suggestions.

Some letters in Figure 1 are too small to read.

Section 4.1. Contribution of EM to understand HIV morphology/maturation has been tremendous. This section could be enriched.

Comments on the Quality of English Language

Better to double-check for grammatical errors. 

Author Response

Thank you for reviewing our manuscript and providing positive comments along with a high evaluation. We sincerely appreciate your time and feedback. Below, we have outlined our responses to your comments.

Comment 1: Some letters in Figure 1 are too small to read.

Response to Comment 1: Thank you for bringing this to our attention. In response to your suggestion, we have increased the font size in Figure 1 for better readability and added additional information to enhance the figure.

Comment 2: Section 4.1. Contribution of EM to understanding HIV morphology/maturation has been tremendous. This section could be enriched.

Response to Comment 2: We have expanded our description of the contribution of EM to HIV maturation research in lines 412-423.